# Georg Simmel Goes Virtual: From 'Philosophy of Landscape' to the Possibilities of Virtual Reality in Landscape Research

**Olaf Kühne** [1,*] **and Dennis Edler** [2]

1   Department of Geosciences, Eberhard Karls University Tübingen, Rümelinstraße 19-23, D-72070 Tübingen, Germany
2   Department of Geography, Ruhr University Bochum, D-44801 Bochum, Germany
*   Correspondence: olaf.kuehne@uni-tuebingen.de

**Abstract:** With his text "Philosophy of Landscape" (German original: "Philosophie der Landschaft"), the German sociologist and philosopher Georg Simmel laid a foundation for landscape research that is still significant today. In the text, he equates the creation and perception of landscape with the creation of a painting. In doing so, he provided an essential foundation for landscape research with a constructivist orientation. In order to be able to grasp the differentiated nature of landscape analytically and to apply it to Simmel's understanding of landscape, we resort to the approach of the three landscapes, which was developed from Karl Popper's theory of the three worlds. The pictorial metaphor of Simmel's understanding of landscapes, however, had the effect of limiting landscape to the visual, and often to what he described as 'natural'. It did not address the power-bound nature of landscape. These aspects, however, are of great importance in current discussions about landscape. Aspects of power, multisensuality, and the incorporation of non-natural elements gain additional currency through the creation of augmented and virtual landscapes. This concerns, on the one hand, the creation of these landscapes, on the other hand, their individual internal consciousness, as well as their social construction. These show, not least, the contingency of landscape construction. They offer possibilities for the investigation of landscape stereotypes, and how innovations can be fed into the social construction of landscape to engage other senses beyond the sense of sight. The aim of our paper is to use conceptual critique to reflect on the conceptual development of social and cultural studies in landscape research since Simmel and to present its potential for framing research on AR and VR landscapes.

**Keywords:** Georg Simmel; virtual landscape; augmented landscape; three landscapes theory; contingency; social constructivist landscape theory

## 1. Introduction

Georg Simmel (1858–1918) is considered one of the founders of (German-language) sociology. In his work, he not only provides constitutive contributions to the sociology of culture, the sociology of cities, the sociology of conflict, and the sociology of emotions [1–4]. He also contributed to the social scientific study of landscapes [5–7]. In the process, according to Lichtblau [3], Simmel did not so much develop a closed theory; his work was more comparable to that of a collector. The fact that collections make it easier to deal with inconsistencies, while closed theories make them more difficult, is also evident in Simmel's remarks on the subject of landscape [8–12]. They are (as will be shown) by no means consistent and closed. In examining them, we follow Gerhard Hard's [13] call to also examine what the remarks do not contain, even though they could contain it. It is precisely this investigation of the contingencies of Georg Simmel's work, that is, of what is neither compelling nor impossible. This leads us to the current questions of landscape research. We investigate how the creation of virtual reality offers new possibilities for experiencing and exploring contingent landscapes. Contingency is a philosophical concept which means, on the one hand, the possibility that something

occurs or does not occur. On the other hand, it means the possibility that something could be completely different from what it is [14]. In the context of landscape, this means: We first deal with the questions of what may be called landscape socially without loss of social recognition—and what may not. The patterns of this evaluation go back, with Simmel, not least, to the ideas of landscape generated by landscape painting. We then broaden this view to the extent to which the binding patterns of landscape construction have differentiated today, i.e., which have become more contingent. We further address the question of which differentiations in landscape understandings emerge as a result of the proliferation of virtual and augmented realities.

There have been notable studies on the systematizations, meanings, and potentials of virtual (and augmented) realities for landscape research [15–20]. However, the topic of multisensory technology receives a rather marginal appreciation. We will focus on this topic—in recourse to and extension of Simmel. Our article draws on current discussions about the theoretical framing of landscape and landscape research, although this is not systematically presented (given our thematic focus). For this purpose, references to existing survey works are given, with different emphases (such as in these publications [21–26]). We complement the theoretical access to landscape with the theory of three landscapes, derived from Karl Popper's theory of three worlds. This addition serves to specify the different levels of understanding of landscape. This operationalization of the three worlds theory assumes that landscape can be divided into three levels: the material level is 'Landscape 1'; the ideas of landscape that are inherent to the individual consciousness is 'Landscape 2'; and the socially shared ideas of landscape refer to 'Landscape 3'. The approach of the three landscapes has proven itself in the past not only to deal with the differentiation of landscape into these levels, but also with the question of the relationships between the levels, as well as the question of which theoretical approaches should be chosen and combined for specific questions in landscape research [27–31]. The theory of the three landscapes for us is an analytical approach. With its help, we see ourselves in a position, firstly, to examine the differentiation of social understandings of landscape, individual attitudes to landscape, and the physical aspects of landscape; secondly, we are in a position to address the alternating influences of Landscapes 2 and 3 as well as Landscapes 1 and 2. The theory of the three landscapes, thus, enables us to present and interpret the different levels of landscapes with their interrelations, interdependencies, and developments. The theory of the three landscapes developed on the basis of the three worlds theory took place in the first decade of the 21st century in Germany and was the basis of numerous studies in which theoretical considerations on landscape were combined with different empirical approaches, which also concerns the reflection of virtual and augmented realities (for example, see [27,32–36]).

With our work, we refer to a specifically German-speaking tradition and show the relevance of this tradition of landscape research, which has existed for eleven decades, on the one hand for international research, and on the other hand for the most current thematic complexes, such as the exploration of landscape in augmented and virtual realities. A connection to international discussions on the social constructedness of landscape (e.g., in [37–39]) remains rather cursory, since on the one hand, a detailed discussion of theoretical approaches is already available in different collections and languages (e.g., [21,23,24,26,40,41]), on the other hand, this would also dilute the deliberately set focus.

This paper reflects on the development of the social science/human geography understanding of landscape, starting from one of the classic works of German-language landscape research. To illustrate the topicality of Simmel's approach, we relate it to one of the most current objects of landscape research, namely augmented and virtual realities. The paper, thus, follows the philosophical tradition of conceptual critique, with the aim of clarifying conceptual access and elaborating a theoretical framework for empirical research, it does not itself present the results of empirical research. This means that by a critique in the tradition of Immanuel Kant [42] we understand an intensive examination of a topic, not in the sense of a neo-Marxist critique, the deconstruction of social relations.



In this paper, Georg Simmel's understanding of landscape is first presented and discussed, especially with regard to multisensuality. After a short summary of current developments in landscape construction with modern geospatial data resources and representation techniques of virtual and augmented reality, the potentials of landscape research are derived. These build theoretically on Karl Popper's three worlds theory and Georg Simmel's philosophy of landscape. In the conclusion, the importance of Simmel for current landscape research is emphasized, and the potentials of VR and AR following Simmel are highlighted.

## 2. The Understanding of Landscape by Georg Simmel

An examination of Simmel's philosophy and sociology in relation to landscape seems worthwhile, even well over one hundred years after its first publication. There are three main reasons why the multisensory foundations of landscape can be coupled with the use of VR:

1. Simmel, in his 1913 work "Philosophy of Landscape" (German original: "Philosophie der Landschaft") [11], provides a basis for a paradigm shift [43] in the social and cultural science of landscape research.
2. Simmel deals not only in his "Sociology of the Senses" (German original: "Soziologie der Sinne") [8], but also in his "Sociology. Investigations on the Forms of Socialization" [9] with the various human senses and their social significance.
3. Simmel's phenomenological approach focuses attention on the experience of spaces as landscapes (more on this in detail can be found in [44]), which provides a perspective beyond cognitivist approaches to the use of VR and AR [20,45].

Georg Simmel's philosophy and sociology are characterized by an oscillation between scientific and feuilletonistic approaches to the world; the texts are often rather essayistic. As a result, Simmel also very often omits references to sources and indications of his method, which makes it difficult to trace the genealogy of his thoughts. This also becomes clear with regard to his remarks on landscape: in his landscape-related texts between the 1910s and the "Philosophy of Landscape" in 1912, he completes the change from an essentialist perspective, which had been dominant in science until then, to a constructivist perspective from today's point of view, without it becoming clear how this came about.

In his "Philosophy of Landscape", Simmel [11] treads a path of landscape research that departs from the prevailing ontologizing and essentializing paradigm of his time (see, for instance, in [46–49]). His work also represents a break with his earlier preoccupations with landscape (for instance, in [10,12]), where he argues, following a rather traditional landscape paradigm. In the "Philosophy of Landscape", Simmel understands landscape as an aestheticizing synopsis of objects. He traces the emergence of landscape back to painting: "The understanding of our whole problem hinges on the motif: the work of art landscape emerges as the increasing continuation and purification of the process in which landscape—in the sense of ordinary language—emerges for all of us from the mere impression of individual natural things (Figure 1). This is exactly what the artist does: he delimits a piece out of the chaotic flow and endlessness of the immediately given world, grasps and forms it as a unity which now finds its meaning in itself and has cut off the world-connecting threads and tied them back into its own center. This is exactly what we do to a lower, less principled degree, in a fragmentary, border-uncertain way, as soon as we now look at a 'landscape' instead of a meadow and a house and a brook and a train of clouds" [11] (p. 12). In Simmel's philosophy, landscape is, according to Hoppe-Sailer [6] (p. 136), determined "as a fundamentally aesthetic category, since it owes its constitution to a visual operation that itself carries aesthetic qualities." In Simmel's understanding, landscape is not given as an object or an expression of a deeper 'essence' that has emerged through the reciprocal imprint of nature and culture, but rather landscape is the result of a selection of material objects that people view together. The conventions of this synopsis, in turn, have to be learned [50]. Moreover, they are culturally and socially differentiated: "Landscape, we say, comes into being by combining a juxtaposition of natural phenomena spread out on the ground into a special kind of unity" [11] (p. 18). According to Simmel [11], the landscape of the causally thinking scholar is different from that of the religiously feeling

nature worshipper, the teleologically oriented farmer, or the general public. This approach to the social and cultural differentiation of the world is not a central element of the "social construction of reality" by Peter Berger and Thomas Luckmann, who actualize the thinking of Georg Simmel ([51]; cf. [52]). In the social constructivist landscape research based on this theory, this social and cultural differentiation of the construction of landscape is also a central theme (among many, [53–57]).

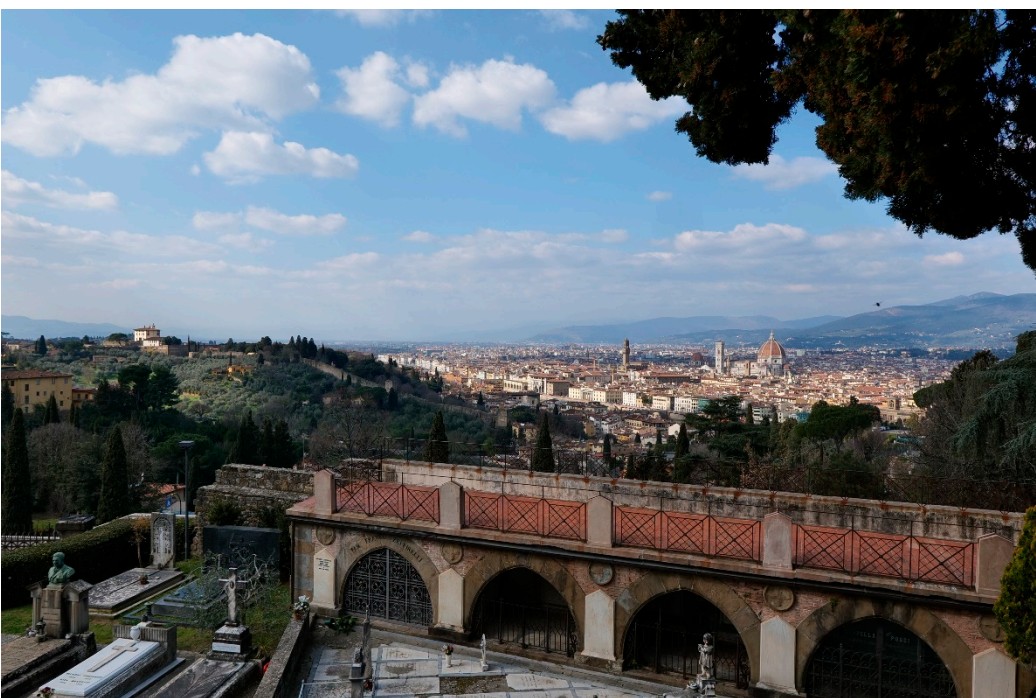

**Figure 1.** The view of Florence today from San Miniato, as it was from more than a century ago, gave Simmel a reason to reflect on the composition of landscape based on the aesthetics of landscape painting. Taking composition patterns borrowed from landscape painting, certain groups of elements (such as blades of grass and flowers to meadow) are aggregated and then subjected to the synthesis "landscape". An essential element of the synthesis to landscape at Simmel's time was (it is partly until today) the attribution of the seen elements as "natural". If objects are clearly of human origin, they are supposed to originate from pre-modern times (Photo: Olaf Kühne 2022).

In (at least) four aspects, current research in landscape theory extends Simmel's theory:

1.  Simmel focuses on the synopsis of objects described as 'natural', although the understanding of 'nature' is again quite broad. Other than trees, hills, and waters, he also includes meadows and grain fields (which would not exist or would only exist for a short time without the intervention of man) as well as houses in his understanding of 'free nature' [11], while he excludes "streets with department stores and automobiles" [11] (p. 7). Current thematizations of old industrial landscapes, urban landscapes, urban land hybrids, etc., integrate these elements. They were not considered by Simmel, in the understanding of landscape (for example, by [58–61]).
2.  An aspect that is not executed in Simmel's focus on visual aspects—as a result of his painting analogy—is that landscape is linguistically conceived and conveyed. This construction of landscape refers on the one hand to the naming of topographic objects (toponyms), but on the other hand also to the linguistic construction of spaces as landscapes [62,63].
3.  Another aspect, and this one is central to our paper, is not found in Simmel's understanding of landscape is the multisensory foundations of landscape. This aspect, too, is omitted in Simmel's work—as a result of the effectiveness of the painting analogy. We will deal with this in the following section.

4.  A central topic of current social and cultural landscape research is not yet reflected in Simmel's understanding of landscape: the topic of power. The power-bound nature of landscape refers to the following questions: On the basis of which power relations is defined, what is to be understood as a preferable landscape? Based on what power relations, who is able to inscribe their needs into physical spaces and how? On the basis of which power constellations are individuals able to innovatively influence landscape concepts? (among many, [64–69]).

The topic of power has been discussed from different theoretical perspectives in landscape research since the 1980s, for instance, on the basis of Marx, Foucault, Bourdieu, Deutsch, and also Popitz, Weber, and Dahrendorf. In our contribution, we follow an understanding derived from the work of the last three authors by understanding power as an opportunity to assert one's will even against opposition, whereby these opportunities can exist from multiple sources (money, political power, technological superiority, definitional sovereignty over what can be stated, etc.; see e.g., [64,70–78]).

## 3. The Multisensuality of Landscape

As shown in the previous section, Simmel's concept of landscape bases is derived from painting. The concept of landscape in Simmel is tied to the visual both on a metaphorical level ('Zusammenschau'—synopsis) and in terms of the perception of objects. This connection is especially remarkable against the background of Simmel's intensive preoccupation with the senses. Such a preoccupation is evident, among other things, in his "Sociology" on the forms of socialization [9] and in the essay "Sociology of the Senses" published in 1907 [8].

Simmel [8] stresses the importance of the eyes for the sensual perception of the world, due to their reciprocity as an organ of reception and expression. Nevertheless, he also refers to the "interaction [...] of eye and ear; for even if neither of them is completely closed to the perceptions of both categories, they are, on the whole, designed for mutual complementation, for the determination of the remaining-plastic essence of man by the eye, and for that of his emerging and sinking expressions by the ear" [8] (w.p.). Simmel sets the sense of smell apart from the senses of sight and hearing. The sense of smell "does not form an object by itself, as face and hearing do, but remains, so to speak, caught in the subject" [8] (w.p.). This is symbolized by the fact that for its distinctions there are no independent, objectively designating expressions, the linguistic version of smells always remains arrested in subjective sensation [8]. Smell in particular can be understood as a 'social sense' in Simmel, thus, it exhibits a particularly 'distinctive' meaning [79]: "The social question is not only an ethical but also a nasal question" [8] (w.p.). Finally, the smell of sweat is connoted with hard physical work or insufficient physical hygiene, i.e., people with a low endowment of 'symbolic capital' [79]. People with a higher endowment of 'symbolic capital' find it easy to remove or cover up body odor that is considered unpleasant (perfume).

Simmel's exclusive focus on the sense of sight in his understanding of landscape as set forth in the "Philosophy of Landscape" is remarkable in other respects as well. The widespread descriptions of spaces at the time, for example in the form of travelogues, descriptions of landscapes and countries, were particularly concerned with acoustic and olfactory stimuli. [80,81]. The focus on the visual is also remarkable against the background of Simmel's phenomenological approach to the world, which would suggest a synesthetic understanding of landscape. The preference of the visual, on the other hand, would be expected in a positivist understanding of landscape, as a result of its quantifiability, whether by measuring dimensions or deriving colors from the visible range of the electromagnetic spectrum [82]. The primacy of the visual arises in Simmel from his direct derivation of 'landscape' from painting. On the one hand, this direct recourse facilitates the tracing back of the 'landscape gaze' to cultural patterns, but on the other hand, it makes the integration of non-visual stimuli into the understanding of landscape more difficult. Here, we see that an established contemporary medium for conveying spatial content channels landscape explorations to the visual dimension. The dominance of the visual also shows great persistence in the scientific approach to landscape. Denis Cosgrove's [37] famous definition

of landscape as less a world that is seen than a way of seeing can also be understood as an indicator of this.

In the recent decades, however, an increasing sensitivity to the non-visual sensory dimensions of landscape has been developed, for instance "soundscapes", "olfactory landscapes", or "smellscapes", but also "foodscapes" have become the subject of research (for instance, in [83–92]). The need to integrate the non-visual dimensions of landscape into research on landscape is also evident, for another reason, landscape is experienced multisensory by those individuals who do not have 'expert special knowledge' [36] about landscape. It is not visually cognized [93].

## 4. Virtual Reality—Some Basic Considerations

In recent years, landscape research has explored the visualization and use of virtually constructed or augmented spaces that go beyond the purely visual dimension, both through geospatial data initiatives and through innovations in the gaming industries. At the policy level, there have been newly established spatial data infrastructures, such as the INSPIRE Directive in Europe. Their main goals lie in opportunities for data exchange and usage, so that spatial data maintained by public authorities is digitally accessible to the broader public [94,95]. In addition, volunteer communities (e.g., OpenStreetMap) acquire and facilitate geospatial data, which significantly expands the resources of publicly available data sources. Through advancing web technologies and computer-based data sharing and communication platforms, private web users also increasingly participate in providing (user-generated) content related to landscape construction, e.g., 3D modeling of meaningful landmarks [96]. Such developments promote opportunities for the customization of digital landscape representations (see, for example, [97–100]). It also supports the creation of new standards to explore and illustrate the smart city represented by the growing technologies of building information modeling (BIM) and digital twins. [101].

Beyond data, the games industry has significantly expanded the possibilities of 3D visualization from the mid-2010s onwards. With the free release of (formerly in-house) game engines, users were given the opportunity to download the software and use it for their own purposes of 3D visualization and landscape construction. Applications can be derived from game engines, and they are compatible with virtual reality hardware systems (e.g., VR headsets). Augmented reality applications can also be created using game engines. Virtual and augmented reality are recent technologies which allow users to experience (immersive) virtual or virtually extended (hybrid) environments. VR is understood as a simultaneous perception and 3D visualization of a simulated environment. AR is the result of a computer-based extension of reality through the superimposition of virtuality.

Modern forms of presentation of virtual and augmented environments in VR support multisensuality. Beyond the visual dimension, acoustic, and also olfactory stimuli can be integrated into VR-based 3D landscapes. The integration of soundscapes and smellscapes into spatial representations has regularly been addressed throughout the development of landscape research and cartographic visualization. In VR-based environments, they can reach a new 'level' of realism in their representation. Both dimensions can be activated at different intensities depending on the properties of the represented locations. Mobile devices (e.g., smartphones, tablets), which are often used for AR applications, as well as VR glasses are equipped with speakers (see [16,102,103]), and first olfactory displays are also VR-compatible (see e.g., [104]). The design of virtual and immersive soundscapes and smellscapes is, thus, technically supported, and landscape visualization can, thus, increasingly expand the purely visual dimension.

Figure 2 shows a section of an animated VR landscape created with the Unity game engine. Spheres around animated objects (moving cars) indicate the range of the spatialized sound representation in 3D. The closer the user (as a virtual avatar) moves to the emitting source, the higher the sound intensity (volume). In an urban traffic situation, this creates an acoustic impression over and above the dominant visual dimension. This can additionally strengthen a spatial sense of immersion.

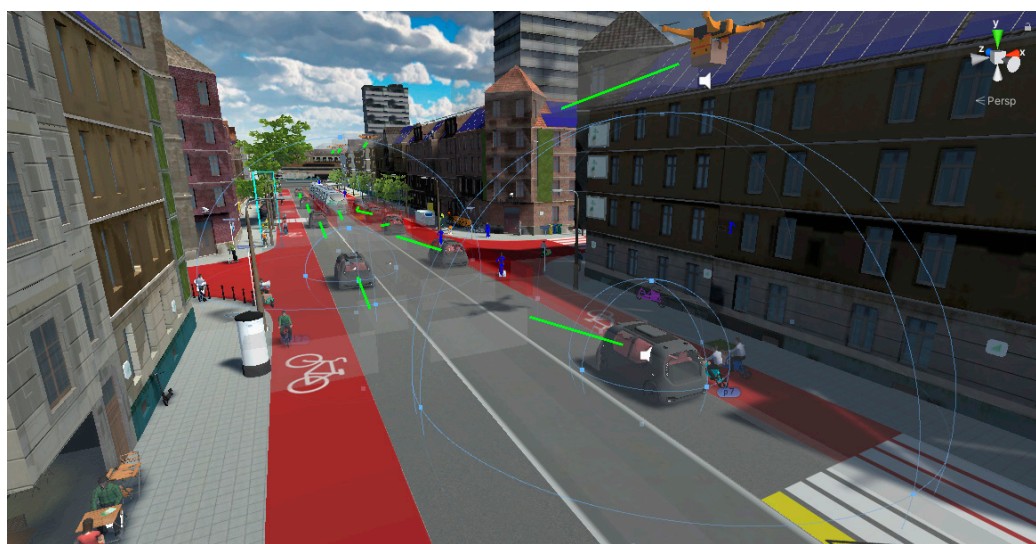

**Figure 2.** Urban soundscape in the game engine Unity (Source: Marco Weißmann and Dennis Edler, Geomatics Group, Ruhr-University Bochum). The sphere around the vehicle indicates the range of the spatial sound intensity in 3D. The level of loudness decreases with a greater distance from the source (vehicle).

## 5. The Virtual and Augmented Construction of Landscape—Based on Karl Popper's Three-World Theory

In Simmel's time, the understanding of landscape in analogy to the work of art was still sufficiently differentiated to be able to communicate scientifically and socially about landscape. However, this has changed in the course of the last approximately 110 years. There are several reasons for this change: (1) Landscape research—following general scientific and scientific-theoretical developments—has produced a variety of understandings of landscape (such as differentiated constructivist, positivist, etc. [21,24,105,106]). (2) In addition, it has become clear—as discussed above—that the reference of humans and physical space in the form of landscape is not solely an aestheticization based on artistic principles, but it is also tangential to other issues, such as the power-laden relationship between individual and social conceptions of landscape. (3) Moreover, the subject matter has been differentiated. Moreover, the subject matter has expanded. The analog world can be translated into a digital world, and the digital world, in turn, can be overlaid (augmented) upon the analog world. This increase in complexity, both on the part of the observer and of the observed, makes an organizing analytical framework necessary. We find this in the theory of the three landscapes, derived from the theory of the three worlds.

In derivation from the three-world theory of Karl Popper [107–109], landscape can be divided into three levels: Landscape 3 comprises the social and cultural understandings of landscape. Landscape 2 comprises the individual approaches to landscape, in particular by updating the contents of Landscape 3 and placing them in one's own lifeworld context. Since this will be of central importance in the following, we will briefly review the terminology of the three, worlds, spaces, and landscapes, here. World 1 comprises the material world, World 2, the world of individual consciousness, and World 3, the world of socially shared knowledge. Space 1 is a part of World 1 in which objects are located in a spatial arrangement. Space 2 characterizes individual ideas of space, and Space 3 is the sum of all socially shared ideas of space. This constitutive level is here, Space 1, without matter in spatial arrangement, there would be neither individual nor social conceptions of space. This is different with landscapes. Here, the constitutive level—as Simmel had already worked out—is the social one, i.e., Landscape 3. The ability to recognize landscapes in spaces is the result of a long history of the formation of conventions. In Germany, this began in the Middle Ages [46,47,110]. Landscape 1 comprises those parts of the material space (Space 1) that are understood as "landscape" by individual consciousness (Landscape

2) and on the basis of social conventions (Landscape 3; among many, [36,111]). To make this clear using Figure 1 as an example, only when I have learned to categorize the objects represented in the photograph (city, hills, trees, sky…), to relate them to each other (e.g., by choosing an elevated view), and to conflate them with social evaluations (e.g., in the sense of Simmel: beautiful, because 'natural' elements are grouped with pre-modern ones), I am able to recognize a Landscape 1 in the material Space 2 depicted here photographically.

Accordingly, these three levels of landscape are interconnected, with Landscape 2 being central, since there is no direct connection between Landscapes 1 and 3. Only Landscape 2 disposes of a body (which is at the same time part of World 1 and can also be attributed to Landscape 1); moreover, Landscape 2 updates the contents of Landscape 3. Landscape 3 affects Landscape 2 by mediating social conventions for constructing land-scapes. Landscape 2 can affect Landscape 3 when landscape innovations are embodied by the individual in social conceptions of landscape. Landscape 2 experiences parts of World 1 as Landscape 1. By means of his body, as part of World 1, ideas of Landscape 2 can physically manifest in Landscape 1 (see Figure 3; to complete what has been said: what has been said here about "landscape" applies analogously to "space" or "world" in general).

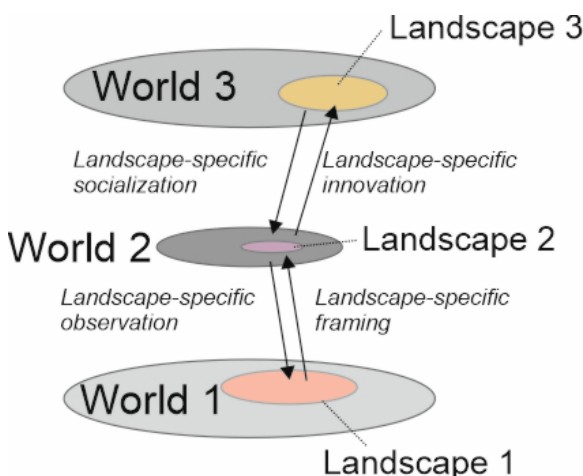

**Figure 3.** Illustration of the relations of Karl Popper's Three Worlds to the three landscapes in the sense of the theory used here, but also of the relations of Landscapes 3 and 2 as well as 2 and 1 (own illustration firstly shown in [36], adapted here).

To illustrate this again with the example of Figure 1, an owner of an area on the hill on the left in the picture is able to intervene in the physical structure of the hill, for example, to clear trees. He succeeds in this because he does not consist of pure consciousness, but of a body that is available to his consciousness. Provided he has learned to operate a saw and has one at his disposal, he is able to clear the trees. What can prevent him from doing so, beyond personal dispositions (such as frailty), are social conventions. These can also be expressed in statutory law, for example, in the prohibition of uprooting healthy trees. However, as far as the legal framework allows, he can strive to create new landscape experiences, for example, by temporarily illuminating trees, decorating them with balloons, etc. With such measures, he can also illustrate the contingency of landscape constructions. He can, thus, have a meta-functional effect, inducing people to clarify conventions about landscape and to reflect on them.

This understanding of landscape needs to be extended if virtual and augmented landscapes are the topics of investigation. According to Kühne [18], this can be clear by prefixing the designation of the layer (1, 2, or 3). These prefix letters are suggested when dealing with landscapes:

- M for material spaces ('real landscape'). This is the kind of landscape that existed before the invention of virtual and augmented realities, such as Simmel's landscape.

That is, the looking in of landscapes into a physical space based on the conventions developed through landscape painting.

- V for virtual spaces. These are virtually constructed landscapes. These include essential landscape construction patterns from the approach of 'traditional' landscape construction, but their basis is not the physical-material space, but the virtually generated space.
- A for augmented spaces. This is a hybridization of the material landscape (M) and the virtual landscape (V) by enriching the landscape M with virtual elements.

The understanding of Landscape 1 as a synthesis of individually actualized (Landscape 2) social ideas (Landscape 3) is, thus, concretized. For example, Landscape V1 denotes the virtual Landscape 1. Landscape A2 denotes the augmented individual construction of landscapes. Landscape M3 denotes the 'traditional' conceptions of landscape). This differentiation results in considerable potential for landscape research [18]. Figure 4 illustrates this in the photo, which was already shown in Figure 1.

The extension of Landscape 1, 2, and 3 by the dimensions V and A means, first of all, an increase in contingency, since new forms of construction, beyond the traditional material ones, are added. This means that the construction of landscape becomes (potentially) more diverse, which expands the research object of landscape research. Dimensions V and A in particular can be specifically incorporated into research on the individual and social construction of landscape. To illustrate this with Figure 4: The possible changes of Landscape M1 are—as explained above—subject to legal restrictions, for example, but also restrictions that are rooted in the inherent laws of World 1 (such as gravity, the upper image in Figure 4). This Landscape M1 can be enriched with elements that do not have to obey these laws (in Landscape A1; middle picture in Figure 4). Instead of hot air balloons, trees could also find their way in. This possibility increases the contingent construction in landscape in the same way as the complete creation of a virtual Landscape V1. This is not bound to the laws of Landscape M1 and is able to sound out the border area of what may still be called landscape and what is already no longer.

Consequently, researchers' construction of Landscape V1 allows to mitigate the complexity of what is represented in order to identify specific patterns of landscape construction. Virtual objects that are to be examined with regard to their significance for the individual and social construction of landscape can be specifically represented. Objects that may have an unplanned influence on the results of experiments can be excluded (e.g., noisy motorcycles or airplanes that may influence the description of Landscape M1 by interviewees). This applies not only to optical stimuli, but also (especially important for the multisensory dimensions of landscape) to sounds, and, if suitably equipped, to tactile and olfactory stimuli.

Thus, the Simmelian restriction of landscape to visual stimuli can be specifically extended: The basic idea of Simmel's landscape theory, the constitutive level of Landscape 3, is maintained. Simmel's approach is extended by the integration of further sensory perceptions. Additionally, the two other 'blind spots' in Simmel can be brought to a scientific investigation on this basis: Landscapes V1 can be consciously shaped. Thus, virtual objects can be integrated, which Simmel did not assign to the 'natural'. Thus, it can be examined whether and how these objects (also with their sounds and potentially smells) are part of the individual (abstracted: social) construction of landscape. Additionally, the isolated representation of stimuli can be used to address questions of the (powerful) relations of Landscape 3 and Landscape 2. For example, issues of the (powerful) socialization of patterns of interpretation, categorization, and evaluation of landscape can be explicitly examined. It is also easier to investigate whether certain objects should not be socially subjected to a different evaluation. Finally, the social and individual understanding of landscape is subject to constant change.

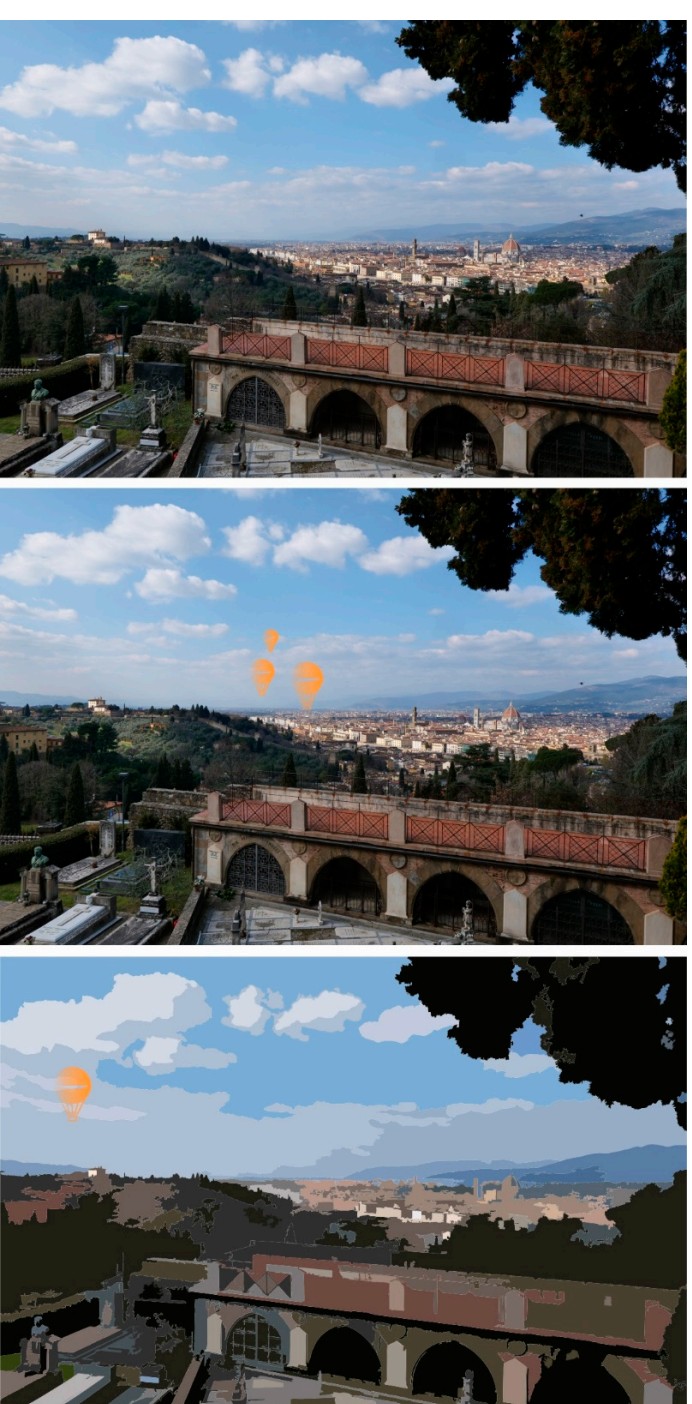

**Figure 4.** Visualization of the differences between the dimensions of M (material), A (augmented), and V (virtual). The basis for the visualization is the assumption that it is not a photo, but an immediate view. Above we see a Landscape M1, i.e., we recognize in a material space, Landscape 1, on the basis of which we learned as a convention (Landscape 3). The middle picture shows Landscape A1, whose M1 basis is enriched with virtual components (hot air balloons). The lower picture shows Landscape V1, as it could occur as a virtual event, e.g., in video games (photo and illustration: Olaf Kühne).

## 6. Discussion

Georg Simmel's paradigm shift from an essentialist understanding of landscape to a constructivist understanding in his "Philosophy of Landscape" can be understood as a decisive step in landscape research. It is remarkable that Simmel himself had previously held an essentialist understanding. It was not until his "Philosophy of Landscape" that

he placed the 'constructedness' of landscape due to social conventions at the center of his considerations. Even if the current social and cultural science landscape research is striving to give more importance to materiality again, Georg Simmel provided a significant contribution to landscape research [112,113]. Moreover, he provided a decisive contribution to overcoming the pre-modern scientific understanding of landscape. Thus, the conceptual distance of Simmel's approach to contemporary social and cultural landscape studies does not lie in its focus on understanding landscape as a function of what is learned, but rather, its distance from topical approaches lies, first, in its blindness to processes of power in the context of the production of Landscapes 1, 2, and 3 (especially interrelationships). Second, it lies in its focus on the visual. This focus, in turn, goes back to his recourse to the creation of landscape in analogy to the image. Even though he has dealt extensively with the sociology of the senses elsewhere, the non-visual does not achieve significance for him in the generation of landscape. Consideration of the multisensory basis of landscape did not receive greater prominence in landscape research until the later second half of the 20th century.

The extension of landscape by an augmented and virtual dimension meant a contingency increase in the social and individual construction of landscape (Landscapes 2 and 3). In order to be able to address these dimensions unambiguously, the designations of Landscapes 1, 2, and 3 are preceded by a prefix letter: A for augmented, V for virtual, and M for the material. This contingency increase is not only structurally conditioned by the extension of the possibilities of production and the experience of landscape in virtual and augmented space, but it also refers to the possibilities of content design. The generation of virtual landscapes enables the creation of worlds beyond the stereotypical landscape expectations of Landscape V1. This concerns both the visual and beyond the visual components [99,114–116]. The generation of Landscape A1 in turn enables the extension of Landscape M1 with additional information. These extensions illustrate the contingent construction of the world in general and landscape in particular. Contingency consciousness is particularly stimulated when the augmented and virtual landscapes are conceived in the spirit of irony (on the spirit of irony in general: [117]; on Landscape A1: [118,119]).

These extensions of landscapes by the dimensions A and V also offer considerable potential for the exploration of the generation of Landscape 2 in its feedback to Landscapes 1 and 3. This can be divided into three aspects: First, stimuli can be used in a controlled manner. This means that the experience of landscape can be focused on the aspects to be explored through the deliberate creation of Landscape V1. Objects and object constellations that are of interest to researchers can be presented, objects that are not of interest can be omitted, and confounding variables are eliminated. Secondly, the confrontation with different Landscape V1 can be conducted in fast motion, so to speak. A wide variety of objects and object constellations can be presented in Landscape V1 in a short time without having to change locations in World 1. This refers, thirdly, to the dimension of multisensory, in that visual stimuli can be combined, in particular, with acoustic, but also tactile or olfactory (see, for instance, 15).

The dimensions of the world represented in the prefix letters show references to each other. With the increasing importance of V3- and A3-landscapes the classical M3-patterns of the construction of the landscape are at least extended. We find this pattern already in the expansion of Simmel's understanding of landscape, which was very strongly (normatively) related to the 'natural' and the 'pre-modern'. It has now expanded to include, for example, industrial-cultural elements. Another effect of the expansion from M-landscape to A- to V-landscape is a differentiation of landscape understandings. While Simmel could still assume that every (educated) person, influenced by landscape painting, saw a similar image in Landscape M1, patterns of interpretation, categorization, and evaluation differ a lot today. This is also due to the fact that the type and intensity of the use of landscapes V1 and A1 are very different.

### 7. Conclusions and Outlook

Through the increase in contingency as a result of the generation of landscapes with the prefixes A and V, the social and individual 'constructedness' of landscapes can be clear. Thus, the generation and use of virtual landscapes also have a great potential to reveal the relationship between power and landscape, which is underrepresented in Simmel's work. This is true not only in academic research, but also in the everyday construction of landscapes [120,121]. Last but not least, patterns of linking visual with non-visual stimuli can be investigated. For example, how stereotypical visual expectations are linked to stereotypical auditory and olfactory expectations, also how dissonances of these expectations are dealt with. In other words, sounds, or smells that are generally not considered to be in harmony with the visual stimuli. Such studies provide insights into the generation and binding nature of stereotypical images of Landscape 3 and their socialization in Landscape 2, but also into the potential of Landscape 2 to innovatively influence Landscape 3.

The "Philosophy of Landscape" by Georg Simmel is still a constitutive basis of landscape research in the social and cultural sciences. In this work, Simmel lays out the basic features of the social construction of landscapes. Building on this core idea, current issues in landscape research can also be addressed. This concerns, for example, the charging of landscape with power, for example, in relation to who is socially granted the right to inscribe their needs in Landscape 1, which aesthetic patterns of interpretation and evaluation become binding in Landscape 3, and which Landscape 2 interpretations can be expressed without loss of social recognition. However, this also concerns the contingencies in the construction of landscapes, for instance through different cultural contexts, disciplines that differ from the stereotypical, or homely patterns of construction. This means that from Space 1, different Landscapes 1 can be synthesized, depending on whether the landscape is constructed in the mode of native attachment, or in a culturally bound common sense, so people of different cultural backgrounds interpret and evaluate landscapes very differently. This shows the contingency of landscape: the synthesis of certain material objects into a landscape is not arbitrary, as it is subjected to a certain aesthetic or ecological interpretations and valuations, but there is also no single universally binding construction of landscapes. However, this also concerns questions of the development of Landscape 2, for instance, in relation to components that are located beyond classical stereotypes (for instance, in the sense of Simmel's natural landscape), such as old industrial objects. The question here is which individuals in which positions of power in society succeed in changing the social construction of landscapes (i.e., innovating from Landscape 2 to Landscape 3). Contributing to the broadening of the scientific understanding of landscape is the integration of the study of non-visual components, as they are common in native ties as well as in the common sense understanding of landscape. As in this case, an increase in landscape contingency can also be observed in relation to current developments in virtual and augmented landscapes.

However, augmented and virtual landscapes are not only able to illustrate the contingency of Landscape 3, but they are also suitable to extend the individual construction of Landscape 2. Regarding landscape V1 or A1, it becomes obvious that, for example, a certain Space 1 would experience a completely different expression under other conditions of use or a changing climate. Thus, by means of everyday world connectivity (i.e., in the modes of native affection as well as common sense), landscapes can be used to illustrate which fundamental side effects human actions can have (such as the emission of greenhouse gases). Such an engagement with the landscape has a high potential for science. The possibilities of controlling stimuli in V1- and A1-landscapes allow us to explore the specifics of the individual and social construction of landscapes, whereby the question of the linguistic version of represented objects is also a worthwhile field of investigation. However, it is not only the possibility of transferring complex scientific results that makes AR and VR relevant for landscape research, but also the investigation of non-scientific landscape constructions (V1 and A1 Landscapes). These can be used in particular to investigate social stereotypes and their reproduction in relation to landscape, research that is still in its

infancy. To return to Simmel, his reflections, which date back eleven decades, still provide valuable interpretations of how a landscape comes into being, namely not that Landscape 1 forms the constitutive level, but Landscape 3 does.

**Author Contributions:** Conceptualization, O.K. and D.E.; methodology, O.K. and D.E.; software, D.E.; validation, O.K. and D.E.; investigation, O.K. and D.E.; resources, O.K. and D.E.; data curation, O.K. and D.E.; writing—original draft preparation, O.K. and D.E.; writing—review and editing, O.K. and D.E.; visualization, D.E.; supervision, O.K.; project administration, O.K. All authors have read and agreed to the published version of the manuscript.

**Funding:** This research received no external funding.

**Institutional Review Board Statement:** Not applicable.

**Informed Consent Statement:** Not applicable.

**Data Availability Statement:** No empirical data was used, as this is a theoretical paper.

**Conflicts of Interest:** The authors declare no conflict of interest.

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
