# Peer review of "Georg Simmel Goes Virtual: From ‘Philosophy of Landscape’ to the Possibilities of Virtual Reality in Landscape Research"

_societies, doi:10.3390/soc12050122_

Round 1

Reviewer 1 Report

This is well referenced and densely argued paper but I have to admit that I found it exceedingly difficult to follow. The author(s) seem to assume that the reader is already familiar with the key philosophical references that they use to build their case. I was not, and soon became badly lost.

There are no issues with the spelling and grammar, but it seems that the authors set out to make every sentence extremely elaborate. Often the complexity is conceptual & philosophical, but there were plenty of sentences where the is purely technical, i.e. stringing together multiple technical terms & acronyms and severely impairing readability -

"At the policy governance level, newly established spatial data infrastructures (e.g., INSPIRE Directive in Europe) with goals of interoperable provision of (official) spatial data led to new data resources."

It might be that the prose has been mangled in translation, or it might be that the philosophical concepts under discussion have been layered and intersected in such a way that renders them almost impenetrable. Overall, it’s very hard to follow the argument being made. And there are  paragraphs, like this one below, for example, which aren't difficult to read, but which simply made no sense to me.

"This leads us across current questions of landscape research to the possibilities that the creation of virtual realities offers for experiencing and exploring contingent landscapes. It means on the one hand the possibility that something occurs or does not occur. On the other hand, it means the possibility that something could be completely different from what it is"

And then there are numerous paragraphs, like this one below, that are so crammed with clauses and cause and effect relationships that I got a headache trying to untangle them:

"Only landscape 2 (world 2) has a body (which is at the same time part of world 1 and can also be attributed to landscape 1); moreover, world/landscape 2 up-dates the contents of world/landscape 3. Landscape 3 affects Landscape 2 by mediating social conventions for constructing landscape; Landscape 2 can affect Landscape 3 when landscape innovations are embodied by the individual in social conceptions of landscape."

I would recommend that the authors imagine that they are speaking to the 'lay reader' in expounding their ideas, or their readership will be very restricted.

Certainly more figures could help too. For example, despite all the mentions of Simmel's interest in landscape paintings, no painting (historical or modern) is shown or described. And the one image that is included, 'an animated VR landscape' could only broadly be termed as such - really it’s a 'streetscape'

Author Response

We sincerely thank you for the constructive review of our manuscript and have incorporated all suggestions, see below:

This is well referenced and densely argued paper but I have to admit that I found it exceedingly difficult to follow. The author(s) seem to assume that the reader is already familiar with the key philosophical references that they use to build their case. I was not, and soon became badly lost.

We admit that our article is written in a very abstract way. Thank you for pointing out that it is more comprehensible. In this respect, we have added examples and explanations.

There are no issues with the spelling and grammar, but it seems that the authors set out to make every sentence extremely elaborate. Often the complexity is conceptual & philosophical, but there were plenty of sentences where the is purely technical, i.e. stringing together multiple technical terms & acronyms and severely impairing readability -

"At the policy governance level, newly established spatial data infrastructures (e.g., INSPIRE Directive in Europe) with goals of interoperable provision of (official) spatial data led to new data resources."

Thank you for pointing us to this problem. In our revision, we tried to re-write the sentences and hope to reduce their complexity in this way.

"At the policy governance level, newly established spatial data infrastructures were established (e.g., INSPIRE Directive in Europe). Their main goals lie in good opportunities of data exchange and usage, so that spatial data ) maintained by public authorities are offered to a broader public and with modern digital options of data accessibility.“

It might be that the prose has been mangled in translation, or it might be that the philosophical concepts under discussion have been layered and intersected in such a way that renders them almost impenetrable. Overall, it’s very hard to follow the argument being made. And there are  paragraphs, like this one below, for example, which aren't difficult to read, but which simply made no sense to me.

"This leads us across current questions of landscape research to the possibilities that the creation of virtual realities offers for experiencing and exploring contingent landscapes. It means on the one hand the possibility that something occurs or does not occur. On the other hand, it means the possibility that something could be completely different from what it is"

We have reworded the paragraph in question as follows:

“This leads us to current questions of landscape research. We investigate the possibilities of the creation of virtual realities offering possibilities for experiencing and exploring contingent landscapes. Contingency is a philosophical concept which means on the one hand the possibility that something occurs or does not occur.”

To clarify out intention, we added:

“In the context of landscape, this means: We first deal with the questions of what may be called landscape socially without loss of social recognition - and what may not. The patterns of this evaluation go back, with Simmel, not least to the ideas of landscape generated by landscape painting. We then broaden this view to the extent to which the binding patterns of landscape construction have differentiated today, i.e. have become more contingent. We further address the question of which differentiations of landscape understandings emerge as a result of the proliferation of virtual and augmented realities.”

And then there are numerous paragraphs, like this one below, that are so crammed with clauses and cause and effect relationships that I got a headache trying to untangle them:

"Only landscape 2 (world 2) has a body (which is at the same time part of world 1 and can also be attributed to landscape 1); moreover, world/landscape 2 up-dates the contents of world/landscape 3. Landscape 3 affects Landscape 2 by mediating social conventions for constructing landscape; Landscape 2 can affect Landscape 3 when landscape innovations are embodied by the individual in social conceptions of landscape."

We realize that the formulation is very abstract, so we have included an example:

“To illustrate this again with the example of figure 1: An owner of an area on the hill on the left in the picture is able to intervene in the physical structure of the hill, for example to clear trees. He succeeds in this because he does not consist of pure consciousness, but a body is available to his consciousness. Provided he has learned to operate a saw and has one at his disposal, he is able to clear the trees. What can prevent him from doing so, beyond personal dispositions (such as frailty), are social conventions. These can also be expressed in statutory law, for example in the prohibition of uprooting healthy trees. However, as far as the legal framework allows, he can strive to create new landscape experiences, for example by temporarily illuminating trees, decorating them with balloons, etc. With such measures, he can also illustrate the contingency of landscape constructions. He can thus have a meta-functional effect, inducing people to clarify conventions about landscape and to reflect on them.”

In addition, we have made the following addition for clarification on another part:

“To make this clear using Figure 1 as an example: Only when I have learned to categorize the objects represented in the photograph (city, hills, trees, sky...), to relate them to each other (e.g. by choosing an elevated view), and to conftone them with social evaluations (e.g. in the sense of Simmel: beautiful, because 'natural' elements are grouped with pre-modern ones), I am able to recognize a landscape in the material space depicted here photographically.”

I would recommend that the authors imagine that they are speaking to the 'lay reader' in expounding their ideas, or their readership will be very restricted.”

Certainly more figures could help too. For example, despite all the mentions of Simmel's interest in landscape paintings, no painting (historical or modern) is shown or described. And the one image that is included, 'an animated VR landscape' could only broadly be termed as such - really it’s a 'streetscape'

When using paintings in publications, there are often copyright problems in Germany. Instead, we have included our own photograph, taken from a vantage point used by Simmel as an example of landscape synthesis.

For this purpose, the following text was added to the caption:

“Figure 1 The view of Florence today from San Miniato, as it was from more than a century ago, gave Simmel reason to reflect on the composition of landscape based on the aesthetics of landscape painting. Based on composition patterns borrowed from landscape painting, certain groups of elements (such as blades of grass and flowers to meadow) are aggregated and then subjected to the synthesis "landscape". An essential element of the synthesis to landscape was at the time of Simmel, it is partly until today, the attribution of the seen elements as "natural", if objects are clearly of human origin, then they are supposed to originate from pre-modern times (Photo: Kühne 2022).”

Reviewer 2 Report

Thank you for giving me the opportunity to review this manuscript. The authors extend Simmel's theory in light of current landscape theoretical research and focus on the multisensuality of landscape. Another contribution of the article is to internationalize the study of the German-speaking tradition.

The theory of three-landscape is not mentioned in the abstract, which is the most important analytical method of the article. The authors should reconsider the structure of the abstract and focus on the purpose, method and conclusion of this study.

Section 1 Introduction: Why choose the theory of three-landscape? I suggest extending the content of the theory so that readers can understand the material without additional reading.

Section 5: The expressions of landscape V1, landscape A2, and landscape M3 are somewhat vague. For example, why does Landscape M3 denote the 'traditional' conceptions of the landscape? The authors need clearly explain the meaning of Landscape M1, M2, M3, V1, V2, V3, A1, A2, and A3. 

Line 363: Why “A for emergent”?

The author mentions that Simmel's third "blind spot" is that "landscape is linguistically conceived and conveyed." But the potential of dimensions A and V in this regard is not addressed in the discussion section, which I suggest can be added.

Author Response

We sincerely thank you for the constructive review of our manuscript and have incorporated all suggestions, see below:

Thank you for giving me the opportunity to review this manuscript. The authors extend Simmel's theory in light of current landscape theoretical research and focus on the multisensuality of landscape. Another contribution of the article is to internationalize the study of the German-speaking tradition.

The theory of three-landscape is not mentioned in the abstract, which is the most important analytical method of the article. The authors should reconsider the structure of the abstract and focus on the purpose, method and conclusion of this study.

Thank you very much for your attentive reading. We have added the following to the abstract:

“In order to be able to grasp the differentiated nature of landscape analytically and to apply it to Simmel's understanding of landscape, we resort to the approach of the three landscapes, which was developed from the theory of the three worlds.”

Section 1 Introduction: Why choose the theory of three-landscape? I suggest extending the content of the theory so that readers can understand the material without additional reading.

We have added the following to the introduction:

“This operationalization of the three worlds theory assumes that landscape can be divided into three levels: the material landscape 1, the landscape 2 as those ideas of landscape that are inherent to the individual consciousness, and the landscape 3, i.e. the socially shared ideas of landscape. The approach of the three landscapes has proven itself in the past not only to deal with the differentiation of landscape into these levels, but also with the question of the relationships between the levels as well as the question of which theoretical approaches should be chosen and combined for specific questions of landscape research [27–31].”

Section 5: The expressions of landscape V1, landscape A2, and landscape M3 are somewhat vague. For example, why does Landscape M3 denote the 'traditional' conceptions of the landscape? The authors need clearly explain the meaning of Landscape M1, M2, M3, V1, V2, V3, A1, A2, and A3.

To clarify this, we added:

“•           M for material spaces ('real landscape'). This is the kind of landscape that existed before the invention of virtual and augmented realities, such as Simmel's landscape. That is, the looking in of landscape into a physical space based on the conventions of seeing developed through landscape painting.,

  • V for virtual spaces. These are virtually constructed landscapes. These in-clude essential landscape construction patterns from the approach of 'tradi-tional' landscape construction, but their basis is not the physical-material space, but the virtually generated space.,
  • A for augmented spaces. This is a hybridization of the material landscape (M) and the virtual landscape (V) by enriching the landscape M with virtual elements.”

Furthermore, we have also explained the theory in more detail earlier with the example of the newly inserted Figure 1:

“To illustrate this again with the example of figure 1: An owner of an area on the hill on the left in the picture is able to intervene in the physical structure of the hill, for example to clear trees. He succeeds in this because he does not consist of pure consciousness, but a body is available to his consciousness. Provided he has learned to operate a saw and has one at his disposal, he is able to clear the trees. What can prevent him from doing so, beyond personal dispositions (such as frailty), are social conventions. These can also be expressed in statutory law, for example in the prohibition of uprooting healthy trees. However, as far as the legal framework allows, he can strive to create new landscape experiences, for example by temporarily illuminating trees, decorating them with balloons, etc. With such measures, he can also illustrate the contingency of landscape constructions. He can thus have a meta-functional effect, inducing people to clarify conventions about landscape and to reflect on them.”

In addition, we have made the following addition for clarification on another part:

“To make this clear using Figure 1 as an example: Only when I have learned to categorize the objects represented in the photograph (city, hills, trees, sky...), to relate them to each other (e.g. by choosing an elevated view), and to conftone them with social evaluations (e.g. in the sense of Simmel: beautiful, because 'natural' elements are grouped with pre-modern ones), I am able to recognize a landscape in the material space depicted here photographically.”

Line 363: Why “A for emergent”?

Thank you for reading so carefully! It must of course be "augmented".

The author mentions that Simmel's third "blind spot" is that "landscape is linguistically conceived and conveyed." But the potential of dimensions A and V in this regard is not addressed in the discussion section, which I suggest can be added.

To intensify these relations, we added:

“Thereby, the dimensions of world represented in the prefix letters show references to each other. By the increasing importance of V3- and A3-landscapes the classical M3-patterns of the construction of landscape are at least extended. We find this pattern - as shown - already in the expansion of Simmel's understanding of landscape, which was very strongly (normatively) related to the 'natural' and the 'pre-modern', but now expanded to include, for example, industrial-cultural elements. Another effect of the expansion from M-landscape to A- to V-landscape is a differentiation of landscape understandings. While Simmel could still assume that every (educated) person, influenced by landscape painting, saw a similar image in landscape M1, patterns of interpretation, categorization, and evaluation differentiate today. This is not least because the type and intensity of the use of landscapes V1 and A1 are very different.”

Round 2

Reviewer 1 Report

The revised title better suited to the discussion

which was developed from the theory of the three worlds.”

– if this theory is Popper’s then best say so here, otherwise the reader will assume it’s Simmel’s, because Simmel is mentioned both right before and right after this sentence

“These aspects, however, are of great importance in current discussions about landscapeAspects that gain an additional topicality with the creation of augmented and virtual landscapes”
- The second sentence is abstruse

Better to combine the two sentences above into one, like this;

These aspects are of great importance in current discussions about landscape, and also pertinent to the creation of augmented and virtual landscapes

“They also offer possibilities for the investigation of landscape stereotypes, but also of the question of how innovations can be fed into the social construction of landscape, also involving the senses, beyond the sense of sight.”

- There’s to many instances of ‘also’ here, and the sentence is a bit too long. You could state more directly;

They offer possibilities for the investigation of landscape stereotypes, and how innovations can be fed into the social construction of landscape to engage other senses beyond the sense of sight

“Simmel did not so much develop a closed theory, but his work was rather characterized by an approach comparable to that of a collector”
- More directly;
Simmel did not so much develop a closed theory; his work was more comparable to that of a collector.

“In examining them, we follow
 Gerhard Hard's call to also examine what the remarks do not contain, even though they could contain it. It is precisely this examination”
- Too many repeats of ‘examine’ here – swap one or two for synonyms

“We investigate the possibilities of the creation of virtual realities offering possibilities for experiencing and exploring contingent landscapes”
- Repeated instance of possibilities. Better;
We investigate how the creation of virtual
 realities offers new possibilities for experiencing and exploring contingent landscapes

“Up to now there are some studies about systematization…”

- Better;
- There have been notable studies about systematization…

“This operationalization of the three worlds theory assumes that landscape can be divided into three levels: the material landscape 1, the landscape 2 as those ideas of landscape that are inherent to the individual consciousness, and the landscape 3, i.e. the socially shared ideas of landscape”

- You need to be very clear about the terms like landscape 1, landscape 2 & landscape 3 as a lot of your argument rests upon the reader understanding them. And when these are all in lowercase, they come across as arbitrary designations, so best emphasise them when they are first mentioned. To state this more directly;

This operationalization of the three worlds theory assumes that landscape can be divided into three levels: the material level is ‘Landscape 1’; the ideas of landscape that are inherent to the individual consciousness are ‘Landscape 2’; and the socially shared ideas of landscape called ‘Landscape 3’

“It is an analytical access, with the help of which the differentiation of social understandings of landscape, individual attitudes towards landscape as well as the physical foundations of landscape with their connections and developments can be represented and interpreted”

- can you simplify or shorten this sentence? It’s hard to follow

The inclusion of Figure 1 improves this paper markedly; it distils, exemplifies and contextualises what is being discussed in the text

“A central topic of current social and cultural landscape research is not yet reflected in Simmel's understanding of landscape: the topic of power. The power-bound nature of landscape refers to the following questions: On the basis of which power relations is defined what is to be understood as a preferable landscape? Based on what power relations, who is able to inscribe their needs into physical spaces and how? On the basis of which power constellations are individuals able to innovatively influence landscape concepts? (Among many: 62– 186 67).”

-
The mention of ‘power’ comes up suddenly here is much too vague. Power is a loaded word and needs to be defined; whose power are we talking about in the landscape? Is it the rich guy living in the castle on the hill, the artist painting the picture, or the intellectual devising the theory? Or is it the power relations inherent in the feudal or industrialised society shaping the landscape?

Can you give an example to make this more concrete?

“Multisensory representation and mediation potentials in landscape painting are limited to visual forms of presentation.

- Probably delete, this repeats what is already explained above

“In the last decades”
- Suggests that the world is about to end! Which may be true, but still, better:
 ‘in recent decades’

“but also "foodscapes" become the subject of research”
- but also "foodscapes" have become the subject of research

“In recent years, landscape research received new impulses for the visualization and use of virtually constructed or augmented spaces that go beyond the purely visual dimension, both through various geospatial data initiatives and through innovations from the multimedia and gaming industries. At the policy governance level, newly established spatial data infrastructures were established (e.g., INSPIRE Directive in Europe). Their main goals lie in good opportunities of data exchange and usage, so that spatial data ) maintained by public authorities are offered to a broader public and with modern digital options of data accessibility”

- This paragraph reads better than previously but is still a little long. I’ve suggested an edit here;

In recent years, landscape research has explored the visualization and use of virtually constructed or augmented spaces that go beyond the purely visual dimension, both through geospatial data initiatives and through innovations in the gaming industries. At the policy level, there have been newly established spatial data infrastructures, such as the INSPIRE Directive in Europe. Their main goals lie in opportunities of data exchange and usage, so that spatial data maintained by public authorities is digitally accessible to the broader public

“represent smart city, which is represented”
- Revise repeated words, i.e. represent & represented. Also – either ‘smart cities’ or ‘the smart city’

“AR is the result of a computer-based extension of reality by adding virtuality”
- Better to say;
AR is the result of a computer-based extension of reality through the superimposition of virtuality.

“Modern forms of presentation of virtual and augmented environments in VR support multisensuality. Beyond the visual dimension, acoustic and also olfactory stimuli …”
- You mention smellscapes and soundscapes here – best cover soundscapes first, as they’ve been around much longer, not just in VR but long beforehand in cinema & radio dramas etc, whereas smellscapes are kind of a new frontier.

“indicate the influence of the 3D sound”
- More precisely, they indicate the range or ‘decay’ of spatialised 3D sound

“This can additionally contribute to an increase of immersion.”
- Again, more precisely ,these sounds are spatial, just like the 3D models are – they support a specifically spatial sense of immersion

Figure 2: Add a sentence to explain what is going on in the image exactly, i.e. how the sphere emanating from the vehicle represents the range of an audible sound

“Whereas at the time of Simmel the understanding of landscape in analogy to the work  
of art was still sufficiently differentiated to be able to communicate about landscape scientifically and socially, this has changed in the course of the last 110 years or so.”

- The above sentence is hard to follow. Be more direct, and let the sentences following afterwards deal with complexities, for example:

Simmel’s understanding of landscape in analogy to art was a study of its time, but much has changed in the course of the last 110 years.

“This has several reasons”
- Better, ‘there are several reasons for this change’

“Moreover, the subject matter has differentiated: The analog world was joined by a digital world, both hybridized in the augmented world”

- Not sure if the digital world has ‘joined’ the digital world…Perhaps better rephrased as:
Moreover, the subject matter has expanded: the analog world can be translated into a digital world, and the digital world, in turn, can be overlaid (augmented) upon the analog world

“Landscape 1 comprises those parts of the material world (world 1) that are understood as 'landscape' individually (landscape 2) ”
- This part has undergone substantial revision, and is certainly clearer than before. However I think a lot of the confusion in these terms is the result of intermixing and interchanging the terms ‘Landscape 1’ and ‘world 1’, which becomes more pronounced in the paragraph following

“Accordingly, these three levels of landscape are interconnected, with landscape 2 being central (as was Karl Popper's world 2), since there is no direct connection between
 landscapes 1 and 3. Only landscape 2 (world 2) disposes of a body (which is at the same time part of world 1 and can also be attributed to landscape 1); moreover, world/landscape updates the contents of world/landscape

- I would suggest prefacing your argument by saying something like “we use landscape 1, 2 and 3 as analogies of Popper’s worlds 1, 2 and 3”

And, thereafter, refrain from comparing ‘worlds’ (1,2 &3) and ‘landscapes’ in the same sentence, i.e. when you say; “landscape 2 (world 2)” or when you say “part of world 1 and can also be attributed to landscape 1”. Moving between two definitions (world and landscape) and then shifting between levels makes the argument hard to follow

“To illustrate this again with the example of figure 1”
- This is a welcome illustration of the concept under discussion and makes the argument more clear

“According to Kühne [18], this can be made clear by prefixing the designation of the layer (1, 2 or 3). These prefix letters are suggested when dealing with landscape:"
- ideally these prefix letters (M, V and A) be represented in a simple diagram or figure - could it be something like a Venn Diagram? Or perhaps these prefixes could be consolidated into Figure 3?

“For example, landscape V1 denotes the virtual landscape 1. landscape A2 denotes the augmented individual construction of landscape. Landscape M3 denotes the 'traditional' conceptions of landscape)”
- you’ve lost me here. With a bit of backtracking, I can find out that ‘V1’ corresponds to virtual spaces, and to ‘Landscape 1’, which is material, but then when I’m presented with ‘M3’, I have no idea what I’m supposed to apprehend.

“the designations of landscape 1, 2 and 3 are preceded by a prefix letter: A for augmented, V for virtual, and M for material.”
-I’m sure these designations and interrelationships are clear in the author’s mind(s), but they aren’t at all easy to follow in the text. I would suggest a diagram to make these associations clear and so that the reader is not completely at sea in the ‘Discussion’ section that follows

The final section ‘Conclusion and outlook’ reads well, but could be a little shorter.

Author Response

Dear reviewer,

we thank you very much for the intensive and constructive review of our manuscript. We highly appreciate the efforts made by the reviewer. It was a pleasure to reflect and change our manuscript according to your comments and suggestion. We hope that the current manuscript version is in an acceptable status for publication.

Best regards

D.E. and O.K.

The revised title better suited to the discussion

Thank you!

which was developed from the theory of the three worlds.”

– if this theory is Popper’s then best say so here, otherwise the reader will assume it’s Simmel’s, because Simmel is mentioned both right before and right after this sentence

Thanks for the advice, we have implemented so.

“These aspects, however, are of great importance in current discussions about landscapeAspects that gain an additional topicality with the creation of augmented and virtual landscapes”
- The second sentence is abstruse

Better to combine the two sentences above into one, like this;

These aspects are of great importance in current discussions about landscape, and also pertinent to the creation of augmented and virtual landscapes

Due to its central importance, we have elaborated on this in more detail:

“Aspects of power, multisensuality, and the incorporation of non-natural elements gain additional currency through the creation of augmented and virtual landscapes. This concerns on the one hand the creation of these landscapes, on the other hand also their individual consciousness-internal as well as the social construction.”

“They also offer possibilities for the investigation of landscape stereotypes, but also of the question of how innovations can be fed into the social construction of landscape, also involving the senses, beyond the sense of sight.”

- There’s to many instances of ‘also’ here, and the sentence is a bit too long. You could state more directly;

They offer possibilities for the investigation of landscape stereotypes, and how innovations can be fed into the social construction of landscape to engage other senses beyond the sense of sight

Thank you very much for the suggestion, we have adopted it as it is.

“Simmel did not so much develop a closed theory, but his work was rather characterized by an approach comparable to that of a collector”
- More directly;
Simmel did not so much develop a closed theory; his work was more comparable to that of a collector.

Thank you very much for the suggestion, we have adopted it as it is.

“In examining them, we follow Gerhard Hard's call to also examine what the remarks do not contain, even though they could contain it. It is precisely this examination”
- Too many repeats of ‘examine’ here – swap one or two for synonyms

In the second sentence we replaced it by “investigation”.

“It is precisely this investigation of the contingencies of Georg Simmel's work, that is, of what is neither compelling nor impossible”

“We investigate the possibilities of the creation of virtual realities offering possibilities for experiencing and exploring contingent landscapes”
- Repeated instance of possibilities. Better;
We investigate how the creation of virtual realities offers new possibilities for experiencing and exploring contingent landscapes

Thank you very much for the suggestion, we have adopted it as it is.

“Up to now there are some studies about systematization…”

- Better;
- There have been notable studies about systematization…

Thank you very much for the suggestion, we have adopted it as it is.

“This operationalization of the three worlds theory assumes that landscape can be divided into three levels: the material landscape 1, the landscape 2 as those ideas of landscape that are inherent to the individual consciousness, and the landscape 3, i.e. the socially shared ideas of landscape”

- You need to be very clear about the terms like landscape 1, landscape 2 & landscape 3 as a lot of your argument rests upon the reader understanding them. And when these are all in lowercase, they come across as arbitrary designations, so best emphasise them when they are first mentioned. To state this more directly;

This operationalization of the three worlds theory assumes that landscape can be divided into three levels: the material level is ‘Landscape 1’; the ideas of landscape that are inherent to the individual consciousness are ‘Landscape 2’; and the socially shared ideas of landscape called ‘Landscape 3’.

We have also adopted this suggestion, thank you!

“It is an analytical access, with the help of which the differentiation of social understandings of landscape, individual attitudes towards landscape as well as the physical foundations of landscape with their connections and developments can be represented and interpreted”

- can you simplify or shorten this sentence? It’s hard to follow

In fact, that was a little tightly worded, we have now made the idea more sophisticated:

“The theory of the three landscapes is for us an analytical approach. With its help, we see ourselves in a position, firstly, to examine the differentiation of social understandings of landscape, individual attitudes to landscape, and the physical aspects of landscape; secondly, we are in a position to address the alternating influences of landscape 2 and 3 as well as landscape 1 and 2. The theory of the three landscapes thus enables us to present and interpret the different levels of landscape with their interrelations, interdependencies and developments.”

The inclusion of Figure 1 improves this paper markedly; it distils, exemplifies and contextualises what is being discussed in the text

“A central topic of current social and cultural landscape research is not yet reflected in Simmel's understanding of landscape: the topic of power. The power-bound nature of landscape refers to the following questions: On the basis of which power relations is defined what is to be understood as a preferable landscape? Based on what power relations, who is able to inscribe their needs into physical spaces and how? On the basis of which power constellations are individuals able to innovatively influence landscape concepts? (Among many: 62– 186 67).”

-
The mention of ‘power’ comes up suddenly here is much too vague. Power is a loaded word and needs to be defined; whose power are we talking about in the landscape? Is it the rich guy living in the castle on the hill, the artist painting the picture, or the intellectual devising the theory? Or is it the power relations inherent in the feudal or industrialised society shaping the landscape?

Can you give an example to make this more concrete?

“To clarify this, we added:

The topic of power has been discussed from different theoretical perspectives in landscape research since the 1980s, for instance on the basis of Marx, Foucault, Bour-dieu, Deutsch, but also Popitz, Weber and Dahrendorf. In our contribution, we follow an understanding derived from the work of the last three authors by understanding power as an opportunity to assert one's will even against opposition, whereby these opportuni-ties can exist on multiple sources (money, political power, technological superiority, definitional sovereignty over what can be said, etc.; see e.g. [62,70–78]).”

“Multisensory representation and mediation potentials in landscape painting are limited to visual forms of presentation.

- Probably delete, this repeats what is already explained above

We deleted the sentence.

“In the last decades”
- Suggests that the world is about to end! Which may be true, but still, better:
 ‘in recent decades’

“but also "foodscapes" become the subject of research”
- but also "foodscapes" have become the subject of research

Thank you for the two hints, we have taken them over.

“In recent years, landscape research received new impulses for the visualization and use of virtually constructed or augmented spaces that go beyond the purely visual dimension, both through various geospatial data initiatives and through innovations from the multimedia and gaming industries. At the policy governance level, newly established spatial data infrastructures were established (e.g., INSPIRE Directive in Europe). Their main goals lie in good opportunities of data exchange and usage, so that spatial data ) maintained by public authorities are offered to a broader public and with modern digital options of data accessibility”

- This paragraph reads better than previously but is still a little long. I’ve suggested an edit here;

In recent years, landscape research has explored the visualization and use of virtually constructed or augmented spaces that go beyond the purely visual dimension, both through geospatial data initiatives and through innovations in the gaming industries. At the policy level, there have been newly established spatial data infrastructures, such as the INSPIRE Directive in Europe. Their main goals lie in opportunities of data exchange and usage, so that spatial data maintained by public authorities is digitally accessible to the broader public

We have also adopted this suggestion, thank you!

“represent smart city, which is represented”
- Revise repeated words, i.e. represent & represented. Also – either ‘smart cities’ or ‘the smart city’

We changed the sentence into:

“It also supports the creation of new standards to explore and illustrate the smart city represented by the growing technologies of building information modeling (BIM) and digital twins.”

“AR is the result of a computer-based extension of reality by adding virtuality”
- Better to say;
AR is the result of a computer-based extension of reality through the superimposition of virtuality.

We have also adopted this suggestion, thank you!

“Modern forms of presentation of virtual and augmented environments in VR support multisensuality. Beyond the visual dimension, acoustic and also olfactory stimuli …”
- You mention smellscapes and soundscapes here – best cover soundscapes first, as they’ve been around much longer, not just in VR but long beforehand in cinema & radio dramas etc, whereas smellscapes are kind of a new frontier.

Thank you for pointing us to this unclarity. We hope that the change of the respective paragraph brings more clarity about this aspect:

“Modern forms of presentation of virtual and augmented environments in VR support multisensuality. Beyond the visual dimension, acoustic and also olfactory stimuli can be integrated into VR-based 3D landscapes. The integration of soundscapes and smellscapes into spatial representations has regularly been addressed throughout the development of landscape research and cartographic visualization. In VR-based environments, they can reach a new ‘level’ of realism in their representation. Both dimensions can be activated in different intensities depending on the properties of the represented locations. Mobile devices (e.g. smartphones, tablets), which are often used for AR applications, as well as VR glasses are equipped with speakers (see [16,102,103]), and first olfactory displays are also VR-compatible (see e.g. [104]). The design of virtual and immersive soundscapes and smellscapes is thus technically supported, and landscape visualization can thus increasingly expand the purely visual dimension.”

“indicate the influence of the 3D sound”
- More precisely, they indicate the range or ‘decay’ of spatialised 3D sound

We totally agree that this sentence can be improved thanks to your suggestion. We changed it into the following:

“Figure 2 shows a section of an animated VR landscape created with the Unity game engine. Spheres around animated objects (moving cars) indicate the range of the spatialized sound representation in 3D.“

“This can additionally contribute to an increase of immersion.”
- Again, more precisely ,these sounds are spatial, just like the 3D models are – they support a specifically spatial sense of immersion

Thank you! We changed the sentence into the following:

This can additionally strengthen a spatial sense of immersion.

Figure 2: Add a sentence to explain what is going on in the image exactly, i.e. how the sphere emanating from the vehicle represents the range of an audible sound

We added these two sentences in the caption of figure 2:

The sphere around the vehicle indicates the range of the spatial sound intensity in 3D. The level of loudness decreases with a larger distance from the source (vehicle).

“Whereas at the time of Simmel the understanding of landscape in analogy to the work  
of art was still sufficiently differentiated to be able to communicate about landscape scientifically and socially, this has changed in the course of the last 110 years or so.”

- The above sentence is hard to follow. Be more direct, and let the sentences following afterwards deal with complexities, for example:

Simmel’s understanding of landscape in analogy to art was a study of its time, but much has changed in the course of the last 110 years.

To make our request clearer, we have formulated it as follows:

“In Simmel's time, the understanding of landscape in analogy to the work of art was still sufficiently differentiated to be able to communicate scientifically and socially about landscape. However, this has changed in the course of the last approximately 110 years.”

“This has several reasons”
- Better, ‘there are several reasons for this change’

Thank you, we have adopted it as such.

“Moreover, the subject matter has differentiated: The analog world was joined by a digital world, both hybridized in the augmented world”

- Not sure if the digital world has ‘joined’ the digital world…Perhaps better rephrased as:
Moreover, the subject matter has expanded: the analog world can be translated into a digital world, and the digital world, in turn, can be overlaid (augmented) upon the analog world

Thank you, we replaced the sentence, as you suggested. 

“Landscape 1 comprises those parts of the material world (world 1) that are understood as 'landscape' individually (landscape 2) ”
- This part has undergone substantial revision, and is certainly clearer than before. However I think a lot of the confusion in these terms is the result of intermixing and interchanging the terms ‘Landscape 1’ and ‘world 1’, which becomes more pronounced in the paragraph following.

To make this clearer, we have added some sentences:

“Since this will be of central importance in the following, we will briefly review the ter-minology of the three worlds, spaces and landscapes here: world 1 comprises the materi-al world, World 2 the world of individual consciousness, and world 3 the world of so-cially shared knowledge. Space 1 is that part of world 1 in which objects are located in spatial arrangement. Space 2 characterizes individual ideas of space and space 3 the sum of all socially shared ideas of space. This constitutive level is here space 1, without mat-ter in spatial arrangement, there would be neither individual nor social conceptions of space. This is different with landscape. Here the constitutive level - as Simmel had al-ready worked out - is the social one, i.e. landscape 3. The ability to recognize landscapes in spaces is the result of a long history of the formation of conventions. In Germany, this began in the Middle Ages. Landscape 1 comprises those parts of the material space (space 1) that are understood as "landscape" by individual consciousness (landscape 2) and on the basis of social conventions.”

“Accordingly, these three levels of landscape are interconnected, with landscape 2 being central (as was Karl Popper's world 2), since there is no direct connection between landscapes 1 and 3. Only landscape 2 (world 2) disposes of a body (which is at the same time part of world 1 and can also be attributed to landscape 1); moreover, world/landscape updates the contents of world/landscape”

- I would suggest prefacing your argument by saying something like “we use landscape 1, 2 and 3 as analogies of Popper’s worlds 1, 2 and 3”

And, thereafter, refrain from comparing ‘worlds’ (1,2 &3) and ‘landscapes’ in the same sentence, i.e. when you say; “landscape 2 (world 2)” or when you say “part of world 1 and can also be attributed to landscape 1”. Moving between two definitions (world and landscape) and then shifting between levels makes the argument hard to follow

Yes, that was a little hard to follow. We have now deleted the "world" from the paragraph and added a half-sentence to explain:

“…to complete what has been said: what has been said here about "landscape" applies analogously to "space" or "world" in general”.

“To illustrate this again with the example of figure 1”
- This is a welcome illustration of the concept under discussion and makes the argument more clear

Thank you!

On the next three comments: Thank you for the reference to an illustration. However, we have attached the whole thing to the photo of figure 1, since we have already explained some things here. With figure 4, the topic should now be more understandable:

“Figure 4: Visualization of the differences between the dimensions of M (material), A (augmented) and V (virtual). The basis for the visualization is the assumption that it is not a photo, but an immediate view. Above we see a landscape M1, i.e. we recognize in a material space a landscape 1, on the basis of which was we learned as convention (landscape 3). The middle picture shows a landscape A1, whose M1 basis was enriched with virtual components (hot air balloons). The lower picture shows a land-scape V1, as it could occur as a virtual event, e.g. in video games (a photo and own illustration).”

We added into the Text also:

“To illustrate this with Figure 4: The possible changes of landscape M1 are - as explained above - subject to legal restrictions, for example, but also to restrictions that are rooted in the inherent laws of world 1 (such as gravity, upper image in Figure 4). This landscape M1 can be enriched with elements that do not have to obey these laws (in landscape A1; middle picture in figure 4). Instead of hot air balloons, trees could also find their way in. This possibility increases the contingent construction of landscape in the same way as the complete creation of a virtual landscape V1. This is not bound to the laws of landscape M1 and is able to sound out the border area of what may still be called landscape and what is already no longer.”

“According to Kühne [18]this can be made clear by prefixing the designation of the layer (1, 2 or 3). These prefix letters are suggested when dealing with landscape:"
- ideally these prefix letters (M, V and A) be represented in a simple diagram or figure - could it be something like a Venn Diagram? Or perhaps these prefixes could be consolidated into Figure 3?

“For example, landscape V1 denotes the virtual landscape 1. landscape A2 denotes the augmented individual construction of landscape. Landscape M3 denotes the 'traditional' conceptions of landscape)”
- you’ve lost me here. With a bit of backtracking, I can find out that ‘V1’ corresponds to virtual spaces, and to ‘Landscape 1’, which is material, but then when I’m presented with ‘M3’, I have no idea what I’m supposed to apprehend.

“the designations of landscape 1, 2 and 3 are preceded by a prefix letter: A for augmented, V for virtual, and M for material.”
-I’m sure these designations and interrelationships are clear in the author’s mind(s), but they aren’t at all easy to follow in the text. I would suggest a diagram to make these associations clear and so that the reader is not completely at sea in the ‘Discussion’ section that follows

The final section ‘Conclusion and outlook’ reads well, but could be a little shorter.

We went through it again, but couldn't bring ourselves to cut anything....